# The Binding of Free and Sulfated Androstenone in the Plasma of the Boar

**DOI:** 10.3390/ani11051464

**Published:** 2021-05-20

**Authors:** Christine Bone, E. James Squires

**Affiliations:** Department of Animal Biosciences, University of Guelph, Guelph, ON N1G2W1, Canada; cbone@uoguelph.ca

**Keywords:** pig, boar taint, androstenone, androstenone sulfate, steroid transport, binding affinity

## Abstract

**Simple Summary:**

Boar taint is characterized by an off-odor or off-flavor in heated pork products that is caused by the accumulation of androstenone in the fat. We have previously demonstrated that androstenone is transported to the fat bound by the plasma protein albumin; however, it is unclear if androstenone sulfate, which is more abundant in the circulation, is transported in the same manner and if the transport of androstenone in the plasma influences the degree of accumulation in the fat. In this article, we determined that androstenone sulfate bound minimally in the plasma of the boar and suggested that this may leave it readily available to enter peripheral tissues, such as the fat where it may enzymatically return free androstenone. Additionally, we demonstrated that the binding of androstenone in the plasma varies significantly between boars with high and low concentrations of androstenone in the fat. This suggests that the binding of androstenone to albumin in the plasma affects the transport and distribution of androstenone within the boar.

**Abstract:**

Androstenone circulates in the plasma bound to albumin before accumulating in the fat, resulting in the development of boar taint. Androstenone sulfate is more abundant in the circulation than free androstenone; however, it is unclear how androstenone sulfate is transported in the plasma and if steroid transport affects the development of boar taint. Therefore, the purpose of this study was to characterize the binding of androstenone sulfate in boar plasma and determine if variability in steroid binding affects the accumulation of androstenone in the fat. [^3^H]-androstenone sulfate was incubated with plasma and the steroid binding was quantified using gel filtration chromatography. Inter-animal variability was assessed by quantifying androstenone binding specificity in plasma obtained from boars that had high or low fat androstenone concentrations at slaughter. Androstenone sulfate bound minimally in the plasma and to isolated albumin, which suggests that it is transported primarily in solution. The specific binding of androstenone quantified in plasma and isolated albumin from low fat androstenone animals was significantly higher (*p* = 0.01) than in high fat androstenone boars. These results indicate that the binding of androstenone to albumin varies amongst individual animals and affects the transport of androstenone in the plasma and accumulation in the fat of the boar.

## 1. Introduction

Boar taint is a meat quality issue characterized by an off-odor or off-flavor in pork products from entire male pigs, which is caused in part by the accumulation of androstenone (5α-androst-16-en-3-one), a 16-androstene steroid in the adipose tissue [1]. Androstenone is produced alongside androgens and estrogens in the Leydig cells of the testes during steroidogenesis at the onset of puberty; therefore, in the swine industry, males are typically castrated early in life to prevent the later development of boar taint [2]. However, castration is recognized as a welfare concern and inhibits androgen and estrogen synthesis, resulting in reduced lean growth and feed utilization [3]. Therefore, an alternative method to castration would improve animal welfare and enhance both the profitability and sustainability of swine production [4].

Following synthesis in the Leydig cells, a significant proportion of androstenone undergoes sulfoconjugation to produce androstenone sulfate, which comprises approximately 70% of the total androstenone present in the peripheral circulation [5,6]. The transport of androstenone in the plasma is a key physiological process that is responsible for delivering the compound to the adipose tissue, where it subsequently accumulates to cause the development of boar taint. Previously, we have identified albumin as the plasma protein responsible for facilitating the transport of free androstenone in the peripheral circulation [7]. However, the binding of androstenone sulfate in the plasma has yet to be characterized.

In several species, albumin has been identified as the plasma protein responsible for binding sulfated steroids, such as estrone sulfate (E_1_S) [8,9,10]. Equine serum albumin was found to facilitate the transport of 95% of estrogen-3-sulfates in the plasma of stallions, which produce similar concentrations of sulfated steroids as the boar [8]. Additionally, E_1_S and 17β-estradiol sulfate bind to a single non-specific binding site on human serum albumin and compete for binding with androgen sulfates, but not unconjugated steroids [9,10]. The affinity of a steroid for a binding protein in the plasma is inversely related to the rate in which the steroid leaves the circulation, known as the metabolic clearance rate, which affects the volume of distribution of that steroid in the plasma [11]. E_1_S has a lower binding affinity for human serum albumin than dehydroepiandrosterone sulfate (DHEAS) but a significantly greater metabolic clearance rate and volume of distribution, which promotes the active uptake of E_1_S into the peripheral tissues, where it is enzymatically deconjugated to return to estrone [11]. Recently, Laderoute et al. [6] demonstrated that androstenone is directly sulfated, which may allow it to function as a steroid reservoir capable of enzymatically regenerating free androstenone. Therefore, the binding efficiency of both androstenone and androstenone sulfate may directly contribute to the development of boar taint and should inversely reflect fat androstenone concentrations.

The purpose of this experiment was to characterize the binding of androstenone sulfate in porcine plasma. Additionally, the specific binding of androstenone was quantified in the plasma of animals with high fat androstenone (HFA) and low fat androstenone (LFA) concentrations to determine if androstenone binding specificity varies amongst animals with differing degrees of androstenone accumulation in the adipose tissue.

## 2. Materials and Methods

### 2.1. Plasma Samples

Plasma and backfat samples were obtained from 13 crossbred ((Yorkshire × Landrace) × Duroc) boars, which were approximately 5 months of age. Boars were housed with non-littermates in groups of approximately 6 and fed standard starter, grower, and finisher rations, formulated by Flordale Feed Mill Limited, ad libitum. Boars were selected at slaughter based on the concentration of androstenone quantified in the fat. All animals were used in accordance with the guidelines of the Canadian Council of Animal Care and the University of Guelph Animal Care Policy. Boars were transported to the University of Guelph’s Meat Laboratory, where they were rendered unconscious with carbon dioxide and exsanguinated, allowing for blood and backfat samples to be collected. Fat androstenone concentrations were determined using a reverse phase high performance liquid chromatography (HPLC) method previously described by Hansen-Møller [12], where androstenone was extracted from fat samples and derivatized with dansylhydrazine, which enabled subsequent quantification by fluorescence detection. In total, 8 plasma samples were selected from animals with high fat androstenone (HFA, *n* = 4) and low fat androstenone (LFA, *n* = 4) concentrations. Plasma samples were incubated with 10 mg/mL activated charcoal, which was stirred overnight at 4 °C, to strip the plasma of endogenous steroids. The plasma was then centrifuged at 2000× *g* at 20 °C for 20 min and filtered using a 0.2 µm non-pyrogenic syringe filter (Sarstedt, Montreal, QC, Canada). Plasma from the remaining 5 crossbred boars was pooled and used for the assessment of steroid sulfate binding, while the specific binding of androstenone was assessed with plasma obtained from the 8 individual boars.

### 2.2. Buffer and Steroid Preparation

Radiolabeled [^3^H]-androstenone (10 Ci/mmol) was purchased from Moravek Biochemicals (Brea, CA, USA). Radiolabeled [6,7^3^H(N)]-E_1_S (51.8 Ci/mmol) was purchased from PerkinElmer (Boston, MA, USA), and unlabeled androstenone was purchased from Steraloids Inc. (Newport, RI, USA). Steroid stock solutions were prepared in 100% ethanol to produce [^3^H]-androstenone (200 µCi/mL), [6,7^3^H(N)]-E_1_S (25 µCi/mL), and unlabeled androstenone (10 mM). Stock solutions were further diluted to produce working solutions in phosphate buffered saline (PBS) (35.8 mM NaH_2_PO_4_, 60.5 mM Na_2_HPO_4_, 154 mM NaCl, 15.4 mM NaN_3_) with 4.6% ethanol. The unlabeled androstenone solution had a final concentration of 18 nmol/mL, and radiolabeled steroid solutions contained 0.056 µCi/mL.

### 2.3. Synthesis and Purification of [^3^H]-androstenone Sulfate

Radiolabeled androstenone sulfate was prepared using human embryonic kidney (HEK293FT) cells purchased from ATCC (Manassas, VA, USA). The cells were plated at 3 × 10^6^ cells per plate in 100 mm cell culture plates (Fisher Scientific, Toronto, ON, Canada) and grown in Dulbecco’s modified Eagle’s medium, supplemented with 10% fetal bovine serum, 1% L-glutamine, 1% penicillin/streptomycin, 1% non-essential amino acids, 1% sodium pyruvate, and 1% geneticin (PAA Laboratories, Etobicoke, ON, Canada) in a humidified atmosphere at 37 °C and 5% CO_2_. Upon 50–80% confluence, the cells were transfected with a 6 µg/plate of the porcine sulfotransferase SULT2A1 (pSULT2A1) vector. constructed as previously described by Laderoute et al. [13], using lipofectAMINE 2000 (Life Technologies, Carlsbad, CA, USA) in accordance with the manufacturer’s instructions. After 48 h, the transfected cells were treated with radiolabeled [^3^H]-androstenone (9.01 µCi), dissolved in ethanol, and dried to produce a final ethanol concentration of 0.1%. After 24 h, cell culture media was collected and analyzed by reverse phase HPLC by injecting 100 µL aliquots onto a Luna 5µ C18(2) HPLC column (250 × 4.60 mm) purchased from Phenomenex (Torrance, CA, USA). The elution of radiolabeled androstenone sulfate was monitored by a β-RAM model 2 isotope detector (IN/US Systems, Tampa, FL, USA) using an HPLC profile optimized for the elution of sulfated 16-androstene steroids, as previously described by Laderoute et al. [6]. This HPLC profile was used to confirm the conversion of androstenone to androstenone sulfate by the HEK293FT cells, which eluted at 34 min and 21 min, respectively. The media was then subjected to solid phase extraction using Sep-Pak C18 solid-phase chromatography cartridges (Waters, Milford, MA, USA), as previously described by Laderoute et al. [13], to isolate [^3^H]-androstenone sulfate from the free steroid. The sulfated steroid fraction was dried under nitrogen and reconstituted in 100% ethanol, which was subsequently diluted in PBS to produce a working solution of [^3^H]-androstenone sulfate (0.056 µCi/mL, 1% ethanol).

### 2.4. Steroid Binding Analysis

The binding of radiolabeled steroids to proteins in porcine plasma was quantified using an HPLC gel filtration method, previously described by Bone et al. [7]. Briefly, radiolabeled [^3^H]-androstenone diluted in PBS (500 µL) was incubated with pooled porcine plasma diluted in PBS (500 µL, 10 mg/mL) and then analyzed by HPLC and subsequent scintillation counting using a Yarra™ 3 µm SC-4000 LC gel filtration column obtained from Phenomenex (Torrance, CA, USA). Additional incubations with radiolabeled [^3^H]-androstenone sulfate and [6,7^3^H(N)]-E_1_S were prepared and analyzed using this method. Radiolabeled steroids diluted in PBS (500 µL) were also incubated with a commercial preparation of fatty acid free human serum albumin or porcine serum albumin purchased from Fisher Scientific (Toronto, ON, Canada) diluted in PBS (500 µL, 10 mg/mL). A standard of human serum albumin was included as a positive control as it is known to bind E_1_S [11] and therefore was used to ensure the validity of the assay for assessing the binding of sulfated steroids. The porcine serum albumin standard was stripped of fatty acids using a charcoal treatment, previously described by Chen [14], to assess the effect of fatty acids on steroid binding. Briefly, porcine serum albumin dissolved in PBS (100 mg/mL) was incubated with activated charcoal (50 mg/mL) at a pH of 3.0, which was stirred on ice for 1 h. Charcoal was subsequently removed by centrifugation at 20,000× *g* for 20 min at 2 °C and the pH was brought to 7.0. All incubations were run in triplicate. Steroid binding was expressed as the percentage of steroid that bound in incubations containing either pooled porcine plasma or albumin standards and was calculated from the ratio of bound to unbound steroid.

### 2.5. Determination of Specific Steroid Binding

The specific binding of androstenone was assessed by quantifying the displacement of [^3^H]-androstenone in a competition assay, which utilized excess unlabeled androstenone as a competitor. The specific binding of androstenone in the plasma was expressed as the percentage of [^3^H]-androstenone that bound per 10 mg of protein, which could be displaced by excess unlabeled androstenone. This was calculated from the ratio of [^3^H]-androstenone that bound in the presence or absence of unlabeled androstenone, which was determined by quantifying the area under the binding curve from two separate experiments (Figure 1). The protein concentration in the plasma from the 8 individual boars was determined using the Bradford assay [15]. The plasma samples were then diluted in PBS to 20 mg/mL. [^3^H]-androstenone was diluted in PBS (500 µL) and incubated with plasma from the 8 individual boars diluted in PBS (500 µL) in the presence or absence of unlabeled androstenone (9 nmol/mL). All incubations were run in duplicate and analyzed by HPLC gel filtration.

Affinity chromatography was used to isolate albumin from pooled porcine plasma, which enabled the direct binding of radiolabeled steroids to porcine albumin to be quantified. Affinity chromatography was conducted using a HiTrap Blue HP albumin-specific affinity chromatography column purchased from GE Healthcare (Mississauga, ON, Canada), as previously described by Bone et al. [7]. Briefly, Plasma was diluted to 20 mg/mL in a 20 mM sodium phosphate binding buffer (pH 7.0) and applied to the column. The albumin fraction was eluted from the column using a 20 mM sodium phosphate and a 2 M NaCl (pH 7.0) elution buffer and was subsequently applied to a PD-10 desalting column purchased from GE Healthcare (Mississauga, ON, Canada) and eluted with Millipore water. Desalted albumin fractions were reconstituted in PBS buffer after freeze drying, and the concentration of albumin isolated from the plasma was determined using the Bradford assay [15]. The isolated albumin (200 µL, 10 mg/mL) was incubated with radiolabeled steroid solutions of androstenone, androstenone sulfate, or E_1_S (200 µL, 0.028 µCi) and analyzed by HPLC gel filtration to evaluate direct steroid binding. All incubations were run in duplicate.

Affinity chromatography was also used to isolate albumin from 8 individual plasma samples belonging to animals with high or low fat androstenone concentrations to assess the specific binding of androstenone to equal concentrations of albumin across individual animals. The specific binding of androstenone to albumin was quantified, as previously described, and was expressed as the percentage of [^3^H]-androstenone that bound per 7.5 mg of albumin. Albumin protein fractions were diluted in PBS (250 µL, 7.5 mg/mL) and incubated with [^3^H]-androstenone (250 µL, 0.028 µCi) in the presence or absence of excess unlabeled androstenone (9 nmol/mL). The binding specificity of [^3^H]-androstenone to albumin was analyzed by HPLC gel filtration. All incubations were run in duplicate.

### 2.6. Statistical Analysis

Statistical analysis was conducted using SAS 9.4 (SAS Institute, Cary, NC, USA). Differences in steroid binding to proteins in the porcine plasma, isolated porcine albumin, or human albumin were evaluated with a one-way ANOVA using the following model with a statistical significance level of *p* < 0.05:y_ij_ = µ + τ_i_ + ε_ij_(1)
where y_ij_ is the percentage of bound steroid; µ is the overall mean; τ_i_ is the fixed effect of the protein treatments; and ε_ij_ is the experimental error, which has a mean of 0 and a variance of σ^2^. A similar model was used to determine significant differences (*p* < 0.05) between steroid treatments in either porcine plasma, isolated porcine albumin, or human albumin, where τ_i_ is the fixed effect of the radiolabeled steroid treatments.

The specific binding of androstenone in plasma obtained from 8 individual boars with either high or low fat androstenone concentrations was analyzed using a two-way ANOVA with a statistical significance level of *p* < 0.05 using the following model: y_ijk_ = µ + τ_i_ + β_j_ +τβ_ij_ + ε_ijk_(2)
where y_ij_ is the specific binding of androstenone in plasma; µ is the overall mean; τ_i_ is the fixed effect of plasma treatments; β_j_ is the block effect of fat androstenone concentrations; τβ_ij_ is the interaction effect between the treatments and blocks; and ε_ij_ is the experimental error, which has a mean of 0 and a variance of σ^2^.

## 3. Results

### 3.1. Binding of Sulfated Steroids to Human Serum Albumin

The binding of radiolabeled steroids to human serum albumin was analyzed by the HPLC gel filtration method to ensure the suitability of this method for assessing the binding of sulfated steroids. All steroids were bound by human serum albumin (Figure 2), and the binding of all steroids to human serum albumin was significantly greater (*p* < 0.01) than the binding observed in porcine plasma and to isolated porcine albumin. The binding of androstenone sulfate and E_1_S to human serum albumin was approximately 3.6- and 6.5-fold greater, respectively, than the binding of these steroids in porcine plasma and porcine albumin (Table 1). There are clear differences in steroid binding between human and porcine albumin that may be in part due to differences in the amino acid composition of the binding pocket.

### 3.2. Binding of Free and Sulfated Steroids in Porcine Plasma and to Isolated Albumin

The binding of radiolabeled steroids in porcine plasma and to isolated albumin is shown in Figure 3 and Figure 4, respectively. Approximately 98% of both androstenone sulfate and E_1_S were unbound in porcine plasma, and the percent binding of steroid sulfates was significantly less (*p* < 0.01) than the percentage of free androstenone that bound. The binding of free androstenone in both porcine plasma and to isolated porcine albumin was approximately 7.8-fold greater than the binding of androstenone sulfate and 5.8-fold greater than the binding of E_1_S (Table 1). Binding of androstenone, E_1_S, and androstenone sulfate to the commercial preparation of porcine albumin that was stripped of fatty acids was 16.3 ± 0.1%, 2.0 ± 0.7%, and 2.7 ± 0.3%, respectively (data not shown). These values were similar to the steroid binding quantified in pooled porcine plasma and isolated porcine albumin.

### 3.3. Assessment of Androstenone Binding Specificity

Plasma was obtained from 8 individual boars, which had either high or low fat androstenone concentrations at slaughter and were incubated with androstenone to assess variability in androstenone binding specificity. Animals with fat androstenone concentrations significantly exceeding the upper boar taint threshold value of 1 µg/g were classified as HFA boars, whereas LFA boars had fat androstenone concentrations close to or less than the threshold of 1 µg/g. Fat androstenone concentrations were quantified by reverse phase HPLC and ranged from 6.73 to 13.72 µg/g in the HFA boars and 0.82 to 1.65 µg/g in the LFA boars. The specific binding of androstenone to plasma proteins in boars with high and low fat androstenone concentrations is presented in Table 2. The specific binding of androstenone in plasma from HFA boars ranged from 7.1 ± 2.4% to 21.9 ± 5.2%, and the specific binding of androstenone in plasma from LFA animals ranged from 18.4 ± 4.6% to 23.4 ± 3.6%. The binding specificity of androstenone quantified in HFA boars was significantly lower (*p* = 0.01) than the binding specificity quantified in LFA boars; however, there was no significant difference in the specific binding of androstenone observed between individual boars within the high and low fat androstenone groups (*p* = 0.14).

### 3.4. Binding Specificity of Androstenone for Isolated Albumin

The average protein concentration quantified in the plasma of HFA boars was significantly greater (*p* = 0.02) than the average plasma protein concentration of LFA animals (Table 2). For this reason, affinity chromatography was used to isolate albumin from the plasma to quantify the binding specificity of androstenone for albumin directly and determine if the inter-animal variability in androstenone binding specificity was a result of differences in the concentration of albumin present in the plasma.

Binding analysis was performed on isolated albumin from the 8 individual HFA and LFA boars. Androstenone was incubated with equal concentrations of albumin isolated from each animal to quantify binding specificity. The average binding specificity of androstenone to albumin isolated from the plasma of HFA and LFA boars is presented in Table 2. The specific binding of androstenone to albumin ranged from 8.4 ± 1.0% to 21.1 ± 2.6% amongst HFA boars. Comparatively, the binding specificity of androstenone to albumin from LFA animals ranged from 19.5 ± 3.6% to 24.3 ± 3.0%. Additionally, there was no significant difference in the binding specificity of androstenone quantified in the plasma and to isolated albumin. The variability in androstenone binding specificity observed between HFA and LFA boars, despite equal concentrations of albumin present in the incubation, indicates that the binding of androstenone to albumin differs between individual animals.

## 4. Discussion

The present study assessed the binding of androstenone sulfate, E_1_S, and free androstenone to proteins in the porcine plasma, purified porcine, and human albumin. Free androstenone bound in pooled porcine plasma to isolated porcine albumin and human serum albumin. In contrast, the binding of androstenone sulfate and E_1_S in both the porcine plasma and to isolated porcine albumin was very low; however, both sulfated steroids bound to human serum albumin. The reduced binding of sulfated androstenone to both human and porcine albumin, relative to free androstenone, may be explained by the inverse relationship that exists between binding affinity and steroid polarity [16]. However, we propose that the differing amino acid composition of porcine and human albumin may have contributed to the dramatic differences in steroid binding observed in this study.

The structure of human serum albumin is predominantly helical and contains three homologous domains (I, II, and III) that are comprised of A and B subdomains [17,18]. The binding of estradiol occurs within the IB domain; however, the existence of a common binding site for all steroid hormones is unlikely [16,19]. Carter el al. [18] reported that the binding of small molecules, such as steroids, to human serum albumin occurs predominantly within the IA and IIIA domains, but the exact location of steroid binding within human serum albumin remains unknown.

The IA (Ser5 to Asp107), IB (Asp108 to Gln196), and IIIA (Pro384 to Tyr497) binding domains of human serum albumin and the corresponding amino acids of porcine albumin are shown in Figure 5 [20]. The binding of steroids to human serum albumin is supported by hydrophobic interactions between the ligand and the inner binding pocket [20,21]. Some regions of human serum albumin, such as the IIIA domain, contain a hydrophobic inner pocket with clusters of polar amino acids towards the outside of the binding domain, which may assist in facilitating the binding of polar steroid sulfates [22]. However, several clusters of polar amino acids within the binding domains of human serum albumin are instead comprised of hydrophobic amino acids in porcine albumin. Conversely, some clusters of hydrophobic amino acids within the human serum albumin binding domains are replaced in pigs by polar amino acids. We hypothesize that differences in the polarity of amino acids between human and porcine albumin may function to impair steroid binding in pigs. Therefore, it is plausible that the reduced steroid binding to porcine albumin that was observed in this study may be caused by amino acids within the binding pocket that impede steroid binding by reducing ligand-binding pocket interactions. Future research should compare differences in the amino acid sequence of albumin between a large number of animals to provide stronger conclusions on the source of inter-animal variability observed in this study.

The limited binding of androstenone sulfate by porcine albumin observed in the present study suggests that albumin preferentially mediates the transport of free androstenone in the boar. However, the Leydig cells produce significantly greater quantities of androstenone sulfate than free androstenone [6,23], but the ability of androstenone sulfate to accumulate in fatty tissues is not known. Androstenone is directly sulfated, and consequently, androstenone sulfate may function as a steroid reservoir [6]. The significant quantity of androstenone sulfate that was unbound in the porcine plasma (~98%) suggests that androstenone sulfate is transported in the circulation of the boar in solution, which may allow the steroid to readily enter peripheral tissues where membrane bound steroid sulfatase enzymes act to deconjugate the steroid, regenerating free androstenone. This theory is in concordance with a previous study that reported an inverse relationship between the binding affinity of E_1_S for human serum albumin and the disappearance of this steroid from the circulation [11]. Future research should investigate the uptake of androstenone sulfate by peripheral tissues, such as the fat, to further characterize the process of boar taint development.

The specific binding of androstenone was investigated by incubating plasma from individual boars that had either HFA or LFA concentrations at slaughter, with radiolabeled androstenone in the presence or absence of an excess unlabeled competitor. The binding specificity of androstenone in animals with HFA concentrations was significantly lower than the binding specificity quantified in the plasma of animals with LFA concentrations. These results suggest that the capacity for specific binding of androstenone in the plasma may influence the accumulation of androstenone in the adipose tissue. However, other unaccounted for factors that regulate the development of boar taint may have influenced the fat androstenone concentrations reported in this study more than the binding specificity of androstenone. The binding specificity of androstenone to albumin that was isolated using affinity chromatography was approximately equal to the binding specificity of androstenone quantified in whole plasma. For this reason, differences in androstenone binding specificity observed may be attributed to inter-animal variability in albumin binding pocket interactions.

Several phosphorylation sites have been identified within the IA and IIIA binding domains of porcine albumin (Figure 5), including Ser5, Ser58, Ser65, Ser418, Thr419, Thr421, and Ser488, which also functions as a ligand binding site [24]. Although it is unclear if albumin undergoes phosphorylation in vivo [26], individual differences in phosphorylation site status could contribute to variability in androstenone binding amongst individuals. Additionally, porcine albumin polymorphisms may have contributed to the inter-animal variability observed in this study. Polymorphisms have commonly been identified in the sequences of serum albumin in cattle, horses, sheep, and dogs [27,28]. Furthermore, canine albumin polymorphisms have been shown to alter the binding of specific drug candidates between individuals [28], which suggests that porcine albumin polymorphisms may have affected androstenone binding amongst individuals in this study. Future research should focus on identifying potential polymorphisms in the sequence of the porcine albumin binding pocket, which correspond with high or low fat androstenone concentrations at slaughter, to determine if albumin is a suitable candidate gene for genetic studies on boar taint.

## 5. Conclusions

This study demonstrated that androstenone sulfate binds minimally to albumin in the porcine plasma, which suggests that it is transported in the circulation, predominantly in solution. These results further support the theory that androstenone sulfate may function as a steroid reservoir in the boar. Furthermore, we have shown, for the first time, variability in the binding specificity of androstenone for albumin among individuals, which is not a result of different plasma protein concentrations and may influence the concentration of androstenone in the fat. This suggests that key differences in the albumin binding pocket may exist between individuals, potentially due to polymorphisms or the degree of phosphorylation, which may affect androstenone binding. Therefore, future studies should investigate the cause of this inter-animal variability to determine if albumin is a suitable candidate gene for boar taint.

## Figures and Tables

**Figure 1 animals-11-01464-f001:**
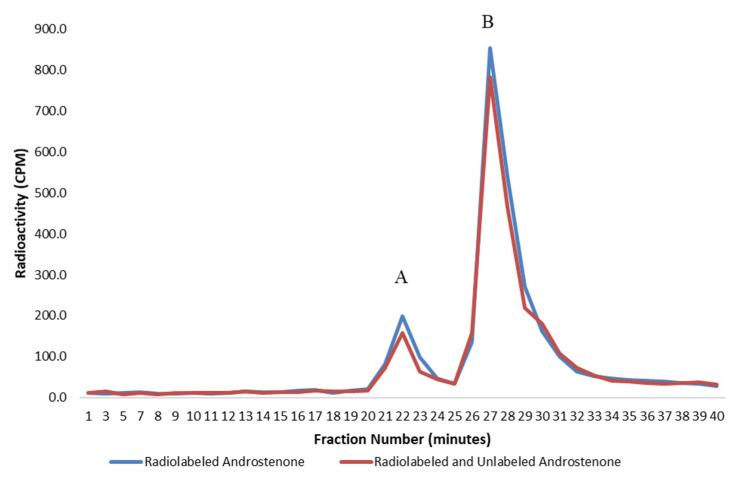
A graph showing the specific binding of radiolabelled androstenone to proteins in porcine plasma determined from the difference in the area under the binding curve, which was quantified from two separate incubations using radiolabeled (blue) or radiolabeled and unlabeled (red) androstenone. The bound and unbound peaks are represented by A and B, respectively. Values are expressed as a mean from two separate experiments.

**Figure 2 animals-11-01464-f002:**
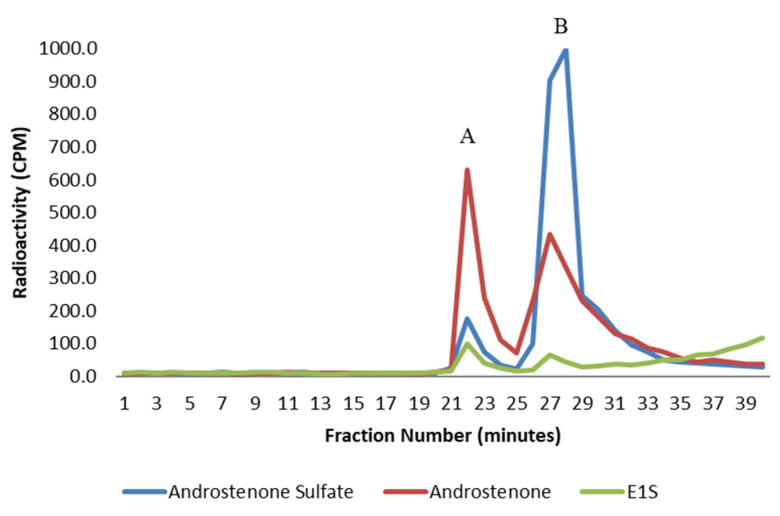
A graph showing the binding of free androstenone (red), androstenone sulfate (blue), and E_1_S (green) to human serum albumin following HPLC gel filtration. Bound and unbound steroid peaks are represented by A and B, respectively. Values are expressed as the mean from three separate experiments.

**Figure 3 animals-11-01464-f003:**
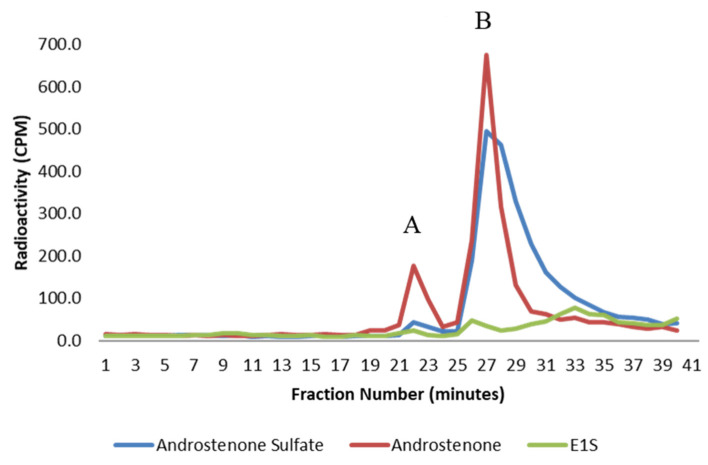
A graph showing the binding of androstenone sulfate (blue), free androstenone (red), and E_1_S (green) in porcine plasma following HPLC gel filtration. Bound and unbound steroid peaks are represented by A and B, respectively. Values are expressed as the mean from three separate experiments.

**Figure 4 animals-11-01464-f004:**
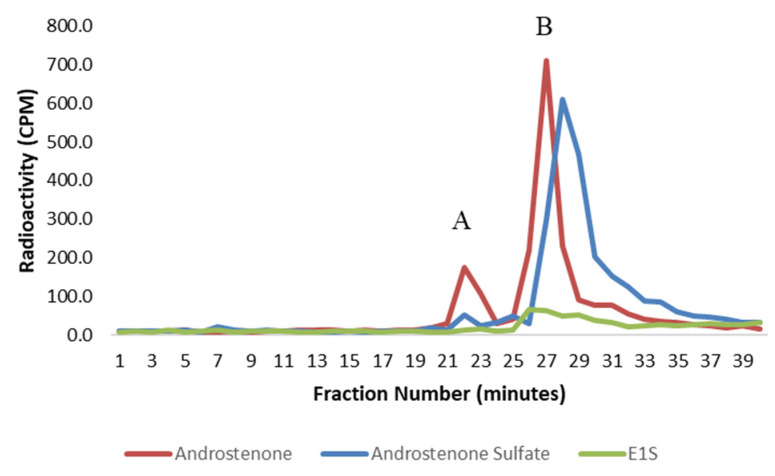
A graph showing the binding of free androstenone (red), androstenone sulfate (blue), and E_1_S (green) to albumin isolated from porcine plasma following HPLC gel filtration. Bound and unbound steroid peaks are represented by A and B, respectively. Values are expressed as the mean from two separate experiments.

**Figure 5 animals-11-01464-f005:**
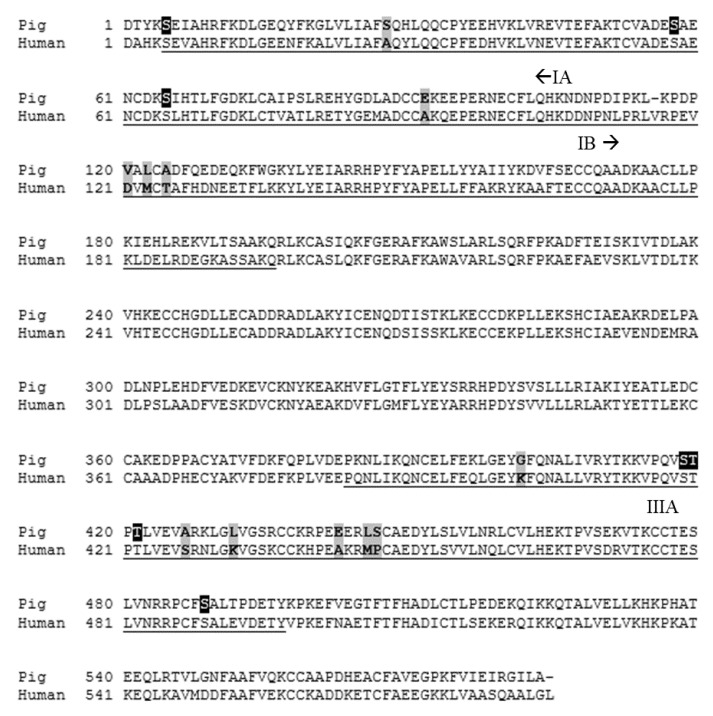
The amino acid sequence of human and porcine albumin [24,25]. Amino acids comprising the IA, IB, and IIIA binding domains of human serum albumin and the corresponding amino acids in pigs are underlined. Key amino acid differences are highlighted in grey, and phosphorylation sites are highlighted in black. Signal sequences have been excluded from the figure.

**Table 1 animals-11-01464-t001:** Binding of Radiolabeled Steroids to Porcine Plasma and Human Albumin.

Radiolabeled Steroid	Percent Steroid Bound (%)		
	Porcine Plasma	Isolated Porcine Albumin	Human Albumin
[3H]-Androstenone	15.1 ± 1.6 ^a^	16.8 ± 0.3 ^a^	34.2 ± 2.0 ^c^
[3H]-Androstenone Sulfate	2.1 ± 0.2 ^b^	2.0 ± 1.2 ^b^	7.5 ± 0.7 ^d^
[6,73H(N)]-Estrone Sulfate	2.6 ± 1.1 ^b^	2.9 ± 1.7 ^b^	17.0 ± 2.1 ^a^

Values are expressed as the mean ± SE; ^a,b,c,d^ Values with a different superscript differ (*p* < 0.05).

**Table 2 animals-11-01464-t002:** The specific binding of androstenone to plasma proteins and albumin and corresponding fat androstenone and plasma albumin concentrations.

	Fat Androstenone (µg/g)	Specific Androstenone Binding to Plasma (%)	Specific Androstenone Binding to Albumin (%)	Plasma Protein (mg/mL)
HFA ^1^	10.7 ± 1.6 ^a^	14.4 ± 3.2 ^a^	14.5 ± 2.8 ^a^	55.7 ± 1.4 ^d^
LFA ^2^	1.1 ± 0.2 ^b^	21.8 ± 1.2 ^c^	22.3 ±1.0 ^c^	50.0 ± 1.2 ^e^

^1^ High fat androstenone boars (*n* = 4). ^2^ Low fat androstenone boars (*n* = 4). Values are expressed as the mean ± SE. ^a,b,c,d,e^ Values with a different superscript differ (*p* < 0.05).

## Data Availability

Not applicable.

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
