# Peer review of "The Binding of Free and Sulfated Androstenone in the Plasma of the Boar"

_animals, 2021, doi:10.3390/ani11051464_

Round 1

Reviewer 1 Report

  • A brief summary

Boar taint is a major problem facing the pork industry and the producers in many countries struggle to find ways of reducing the taint in order to avoid the need for castration of male pigs. This work examines the circulation in the blood of androstenone, one of the major compounds responsible for boar taint. The study evaluated the binding of sulfated form of androstenone to blood plasma protein of boars with the conclusion that androstenone sulfate binds to plasma proteins only marginally, and thus androstenone is predominantly transported in a free form. The authors also studied the specific binding of free androstenone in boars with high vs low androstenone concentrations in the fat. The results showed the difference between those two groups suggesting an effect on androstenone blood transport on boar taint level. Altogether, the study provides new insights into the transport of androstenone in blood and its importance for the development of boar taint.

  • Broad comments 

Main strengths this work include the high quality of the experimental work and the entire manuscript i.e. introduction, methodology, including the statistical analysis, presentation of findings, discussion and conclusions. The main findings are the poor biding of sulfate conjugated androstenone to the blood plasma protein of boars and the differences in the specific biding of androstenone to plasma protein and albumin in male pigs with different levels of androstenone in fat and thus boar taint. Those new and important findings open up new possibilities for explaining the variation on androstenone levels in fat and thus boar taint in entire male pigs.

Minor weaknesses   

Characterization of the animals used. It would be useful to describe the animal material better. As it now, only breed and weight are given. What was the age of the boars? Were they littermates? How were they fed, the diet? Did the authors select those specific animals, what were the criteria? Boar taint level is known to be effected by genetics, diet and other environmental factors. It is important information for the validity of the conclusions and for the repeatability of the study.  

Low number of animals and poor description of the animals used for the experiments on specific binding of androstenone is a weakness of this study and warrants careful conclusions. The authors’ conclusions are appropriately moderate. However, I suggest mentioning that in the discussion section so that the reader understands the limitations of the present study. Binding of small molecules to plasma protein is known to vary to a high degree among individuals.

The technique used for the evaluation of specific binding of androstenone is a valid method; however, binding studies may render different results depending on the method used. Therefore, I suggest that one of the conclusions should be that specific binding of androstenone in boar plasma needs to be further assessed with different methodology.

  • Specific comments

Entire manuscript

The reference numbering is incorrect. It starts already in the Introduction; number 7 should be number 6, I think, and this discrepancy continues throughout the manuscript. There is no reference number 27. Please correct.

 Introduction

Boar taint in entire male pigs is caused by androstenone as well as another compound called skatole. To state that boar taint is primarily caused by androstenone is therefore not entirely correct. I suggest mentioning in the introductory part of the manuscript also skatole even though skatole is not a subject on this study. The risk for boar taint will not be eliminated by targeting androstenone alone as it is suggested in the manuscript.

Materials and Methods

There is no information about the molar concentration of the radiolabeled steroids used in the binding analyses, only radioactivity expressed as Ci/ml. Is that a reason for that? Please include that information i.e. molar concentration of labeled androstenone. Binding activities are often concentration dependent. It is also important for the repeatability of the study.

Line 82-83. I suggest including more information on the animals, see Broad Comments.

Line 97. Delete [(Yorkshire x Landrace) x Duroc] – redundant.

Line 197. “…bound/7.5 mg”. I am not sure if the use of “/” is correct.

Author Response

Thank you for taking the time to review our manuscript. We have addressed each of the comments below (shown in bold) and have made the appropriate changes to the manuscript.

Reviewer 1:

Characterization of the animals used. It would be useful to describe the animal material better. As it now, only breed and weight are given. What was the age of the boars? Were they littermates? How were they fed, the diet? Did the authors select those specific animals, what were the criteria? Boar taint level is known to be effected by genetics, diet and other environmental factors. It is important information for the validity of the conclusions and for the repeatability of the study.  

Low number of animals and poor description of the animals used for the experiments on specific binding of androstenone is a weakness of this study and warrants careful conclusions. The authors’ conclusions are appropriately moderate. However, I suggest mentioning that in the discussion section so that the reader understands the limitations of the present study. Binding of small molecules to plasma protein is known to vary to a high degree among individuals.

We have mentioned this limitation on line 333.

The technique used for the evaluation of specific binding of androstenone is a valid method; however, binding studies may render different results depending on the method used. Therefore, I suggest that one of the conclusions should be that specific binding of androstenone in boar plasma needs to be further assessed with different methodology.

The binding of androstenone in the plasma has been assessed using different methods in the literature. In our original paper where we developed this method we did discuss this point a lot and would like to avoid repeating conclusions from our previous paper. We have already included suggestions on how we feel this work could be further validated (identification of potential polymorphisms in the albumin binding pocket between individuals).

  • Specific comments

Entire manuscript

The reference numbering is incorrect. It starts already in the Introduction; number 7 should be number 6, I think, and this discrepancy continues throughout the manuscript. There is no reference number 27. Please correct.

We have corrected an auto numbering error in the references so the references within the manuscript now properly align with the references at the end.

 Introduction

Boar taint in entire male pigs is caused by androstenone as well as another compound called skatole. To state that boar taint is primarily caused by androstenone is therefore not entirely correct. I suggest mentioning in the introductory part of the manuscript also skatole even though skatole is not a subject on this study. The risk for boar taint will not be eliminated by targeting androstenone alone as it is suggested in the manuscript.

We have changed “primarily caused by androstenone” to “caused in part by androstenone”. We agree that skatole also can contribute to the development of boar taint, but would prefer not to discuss it as androstenone is the focus of the paper.

Materials and Methods

There is no information about the molar concentration of the radiolabeled steroids used in the binding analyses, only radioactivity expressed as Ci/ml. Is that a reason for that? Please include that information i.e. molar concentration of labeled androstenone. Binding activities are often concentration dependent. It is also important for the repeatability of the study.

The reason why we have chosen to express the radioactivity in uCi/ml is because the radiolabeled steroid was only used as a tracer in the binding experiment. The concentration was included for the unlabeled steroids used.

Line 82-83. I suggest including more information on the animals, see Broad Comments.

We have added this information.

Line 97. Delete [(Yorkshire x Landrace) x Duroc] – redundant.

We have deleted this.

Line 197. “…bound/7.5 mg”. I am not sure if the use of “/” is correct.

“/” was changed to “per”. This was also changed on line 162 where it was also used.

Reviewer 2 Report

The study is very well presented and deals with an interesting aspect of transport of androstenone in male pigs. The experimental design is fully detailed and different methods were carried out to better understand the role of albumin. Maybe some precision on the age of animals could be added (only an approximate age is provided).

The point that could be more discussed is the interest of albumin in the transport of androstenone sulfate considering that sulfoconjugation makes compounds water-soluble.

Part of the plasma were pooled to provide a reference, but it might have been more appropriate to use a pool of all the samples.

Some minor comments:

L20: development of boar taint

L150: previously instead of previous

L254: the threshold value (1 µg/g) between high and low seems inadequate, as average value for low animal is 1.1 µg/g. Because HFA boars are more variable, would it be possible to study the binding ability of albumin in that group only ?

The letters in table 2 are unclear. In the title of the last column, is it protein or albumin ?

L342: the steroids are deconjugated when entering adipose tissue, but what happens when steroids go from adipose tissue to blood ? is there a conjugation by the adipose tissue ?

L352: It is contradictory to the fact that was mentioned in the previous paragraph (the inverse relationship between binding ability and disppearance of the steroid) ?

The figure 5 should be put before in the text.

Author Response

Thank you for taking the time to review our manuscript. We have addressed each of the comments below (shown in bold) and have made the appropriate changes to the manuscript.

Reviewer 2:

The point that could be more discussed is the interest of albumin in the transport of androstenone sulfate considering that sulfoconjugation makes compounds water-soluble.

In the introduction we address this on line 56 “In several species, albumin has been identified as the plasma protein responsible for binding sulfated steroids such as estrone sulfate (E1S) [8,9,10].” This study is an extension of our previous study, which characterized a binding protein for free androstenone. We suspect androstenone sulfate may also be involved in the development of boar taint, which is why we attempted to identify a potential binding protein.

Some minor comments:

L20: development of boar taint

We are not sure what the suggestion is. We have left it as is.

L150: previously instead of previous

This has been fixed.

L254: the threshold value (1 µg/g) between high and low seems inadequate, as average value for low animal is 1.1 µg/g. Because HFA boars are more variable, would it be possible to study the binding ability of albumin in that group only ?

The threshold value for boar taint has been established from trained sensory panels. 1ug/g is a sufficiently high quantity of androstenone to detect boar taint in the meat. However, in our study we were interested in the binding differences between animals with very extreme fat androstenone values and animals with levels near the established threshold or below. We believe it is best to compare the groups as they are.

The letters in table 2 are unclear. In the title of the last column, is it protein or albumin ?

The last column is plasma protein concentration.

L342: the steroids are deconjugated when entering adipose tissue, but what happens when steroids go from adipose tissue to blood ? is there a conjugation by the adipose tissue ?

We have preliminary work to suggest that conjugated steroids including androstenone sulfate can enter the fat and subsequently undergo deconjugation. Deconjugation is facilitated within the adipose tissue by sulfatase.

L352: It is contradictory to the fact that was mentioned in the previous paragraph (the inverse relationship between binding ability and disppearance of the steroid) ?

L352 states that androstenone sulfate may be able to accumulate in peripheral tissues such as the fat more readily than free androstenone. This is in agreeance with the inverse relationship between binding affinity and disappearance of the steroid as androstenone sulfate was not bound by albumin. This idea is again supported by our new preliminary work.

The figure 5 should be put before in the text.

Figure 5 is shown before it is mentioned in the text.